

# Learning from conceptual models – a study of emergence of cooperation towards resource protection in a social-ecological system

Saeed Harati-Asl[1], Liliana Perez[2], Roberto Molowny-Horas[3]

[1]Roger Tomlinson laboratory, Department of Geography, McGill University, Montreal, H3A 0B9, Canada;
saeed.haratiasl@mcgill.ca
[2]Laboratory of Environmental Geosimulation (LEDGE), Department of Geography, Université de Montréal, Montreal, H2V 0B3, Canada; l.perez@umontreal.ca
[3]Centre de Recerca Ecològica i Aplicacions Forestals (CREAF), Bellaterra, Cerdanyola de Vallès, E-08193 Spain; roberto@creaf.uab.es

*Correspondence to*: Saeed Harati-Asl (saeed.haratiasl@mcgill.ca)

**Abstract.** Engaging ecological resource users in intervention to protect the resource is challenging for governments due to self-interest of users and uncertainty about intervention consequences. Focusing on a case of forest insect infestations, we addressed questions of resource protection and environmentally responsible behavior promotion with a conceptual model. We coupled a forest infestation model with a social model in which a governing agent applies a mechanism for recognition and

promotion of environmentally responsible behavior among several user agents. We ran the coupled model in various scenarios with a Reinforcement Learning algorithm for the governing agent as well as best-case, worst-case, and random baselines. Results showed that a proper recognition policy facilitates emergence of environmentally responsible behavior. However, ecosystem health may deteriorate due to temporal differences between the social and ecological systems. Our work shows it is possible to gain insight about complexities of social-ecological systems with conceptual models through scenario analysis.

**Keywords:** Social-ecological system, Agent Based Model, Reinforcement Learning, conceptual model, emergence

## 1 Introduction

### 1.1 A governance problem in sustainable development

Sustainable development is defined as a development that provides the needs of the present time, without sacrificing the ability of future generations to provide their needs (Brundtland, 1987). The criteria for such development are known to have social,

ecological, and economic dimensions (Brown et al., 1987; Barbier, 1987). In the present study we are interested in a group of sustainable development problems wherein a government seeks help from users of a natural resource, which is at risk, to protect that resource. From the government's viewpoint it is ideal that the users find the motivation to cooperate with the government. However, such motivation may not come out of the users' sense of altruism (Kaplan, 2000). One possible way to build that sense of cooperation may be to offer a financial incentive, but that may not be enough to create a long-lasting

motivation for environmentally responsible behavior, either (de Young, 2000; Katzev & Johnson, 1987). Moreover, experience





shows that using authority to enforce environmentally responsible behavior may fail (Blundell & Gullison, 2003; Feeny et al., 1990; Wagner, 2004; Wittemyer et al., 2011). Therefore, in the ideal situation in the government's view, users should voluntarily adopt a behavior that implies costs to them, without the governing entity needing to use force or financial incentives. In addition, in that ideal situation, the said behavior successfully protects the natural resource. These considerations give rise

to a question: is such an ideal situation possible?

In this paper, our interest is in a particular type of the above-mentioned sustainability problems, where the ecological system is a forest resource that is attacked by a pest, and the social system includes users of the resource and a governing entity. We assume a case where the users are logging companies and the government wants them to cut specific parts of the forest and create buffer zones to control the spread of the pest. This problem is multi-disciplinary: it involves land change science (Lambin

et al., 2006) in the study of changes in the ecological system; it relates to the domains of collective action (Nyborg et al., 2016; Ostrom, 2009) and social norms (Farrow et al., 2017; Nyborg et al., 2016) in the analysis of formation of a common behavior among users; it entails the field of social-ecological systems (Liu et al., 2007; Ostrom, 2009) in the endeavor to understand the dynamics that emerge through coupling the society with the forest; and, in a broader view, this problem and the question of how to address it are in the realm of complex systems (Cosens et al., 2021; Filotas et al., 2014).

**1.2 Background from multiple disciplines**

Complex systems are entities composed of elements and interactions that make the system behave as a whole, with such characteristics as self organization, non-linearity, emergence, feedback, and path-dependence (O'Sullivan, 2004). Because of these characteristics, the dynamics of complex systems involve novelty and surprise (Batty and Torrens, 2005). This causes a concern in problems of sustainable development, as they typically involve intervention in or experimentation with complex

systems, and particularly social-ecological systems. Due to the uncertainty and complexity that is inherent in these systems, it is not always ethically and logistically justifiable to perform trial-and-error experiments on them (Kriebel et al., 2001), as intervention in these systems may have unanticipated, irreversible and adverse effects. This concern justifies learning by modeling and simulation (Janssen and Ostrom, 2006a).

Societies and ecosystems are complex systems. When a society uses a natural resource, the link between the two systems

creates a larger complex system, referred to as a social-ecological system (Berkes and Folke, 2000). Social-ecological systems (SES) demonstrate complexities that cannot be understood through the lens of sociology or ecology alone (Liu et al., 2007). In a typical SES, the society receives ecosystem services (Daily, 2000; Millennium Ecosystem Assessment, 2003) and makes changes in the ecosystem. To combine social and ecological knowledge in the analysis of the complexity of SES, a framework has been developed, which accounts for governance and resource systems at larger scales, as well as users and resource units

at smaller scales (Ostrom, 2009). This framework has been implicit in sustainability studies in a variety of domains such as sustainable navigation (Parrott et al., 2011), fishery (Schlüter et al., 2014), and forest management (Wimolsakcharoen et al., 2021).





Many studies of complex systems involve building and using models that replicate some aspects of those systems (Railsback & Grimm, 2012; Wolfram, 2002). Simulations of complex systems are often built with a bottom-up approach, using methods

such as Agent-Based Models (ABM) and Cellular Automata (CA) models (Grimm et al., 2005). An ABM is made up of several computer programmed agents that interact with each other and their environment, and act upon their decision rules (Castle and Crooks, 2006). A CA model is composed of a grid of cells, where the state of each cell is defined by a rule based on the previous states of that cell and its neighbors (White and Engelen, 1993). ABMs have been used in a wide variety of complex systems studies, such as epidemiology (Perez and Dragicevic, 2009), animal movement (Bonnell et al., 2013), land

development (Pooyandeh and Marceau, 2013) and forest disturbance (Perez and Dragicevic, 2010; Katan and Perez, 2021). Likewise, CA have been used in research works within fields such as land change (National Research Council, 2014; Lambin and Geist, 2006), urban growth (Batty et al., 1999; Clarke et al., 1997; de Almeida et al., 2003) and forest disturbance (Bone et al., 2006; Gaudreau et al., 2016), among others.

Emergence of behavior in a society is a subject of study in the field of social norms. Several definitions of norms have been

stated in social sciences literature. In one definition, norms are cultural rules that guide people in their behavior (Ross, 1973). In another definition, norms are social rules that govern the encouragement or condemnation of certain behaviors (Savarimuthu and Cranefield, 2011). Norms have also been defined in the context of institutions (Ostrom, 1990; Crawford and Ostrom, 1995) as valuations of actions regardless of the immediate consequences of the actions. Institutions can formalize norms by converting them to regulations (North, 1990). In yet another view, norms are classified as descriptive and injunctive.

Descriptive norms show what others do, whereas injunctive norms show what others approve of (Cialdini et al., 1990). A review of literature on SES governance indicates that social norms largely influence environmentally responsible behavior (Bourceret et al., 2021). Literature also highlights that emergence of environmentally responsible behavior in a society depends on what the individuals do and what they favor (Nyborg et al., 2016), which, by the above definitions, are the equivalents of descriptive and injunctive norms, respectively.

**1.3 Setting, questions and objectives**

In this study we are interested in a SES governance problem. We consider a setting where the government needs the participation of users of a forest in a management action with the aim of protecting the forest against infestation outbreaks. To clarify the scope of the problem we state the following assumptions:

- The forest is at risk, and the state of health of the forest urges the government to act towards its protection.

- The expected action to protect the forest can only be performed by the users but is costly for them.

- The users are driven by self-interest, and not by altruism.

- The government cannot offer financial incentives for the purpose of enticing the cooperation of the users.

- The government cannot enforce its authority and oblige the users to cooperate with it.





- Users have a desire for good reputation, which motivates them to perform environmentally responsible behavior. However, there is no social sanction or punishment for individuals who do not demonstrate responsible behavior.

- The government's knowledge of the social system is limited. It does not know the decision criteria of the users.

- The government's knowledge of the ecosystem is limited. Although the government wishes to intervene and protect the forest, the government is not certain about the consequences of its desired intervention.

In previous works we developed an ecological model of outbreaks of a forest insect infestation (Harati et al., 2020, 2021b) and a social model of promotion of a new behavior norm (Harati et al., 2021a). In the present work we combine the above two models to gain insight about a particular intervention measure to control the spread of infestations. The intended measure is to encourage forest users to voluntarily create buffer zones in vicinity of newly observed infestations.

In a SES, the ecological and social systems are each the variable environment of the other. This matter creates complexities, which, in the particular case of our study, give rise to the following questions: If forest users fully cooperate with the governing entity, will creation of buffer zones be effective in suppressing the spread of infestations in the forest? Will forest users cooperate with the governing entity? In case of partial cooperation of users with the governing entity, how will the infestations spread in the forest? Considering these questions, the objectives of this study are:

- To build a SES model by coupling the above mentioned social and ecological models;

- To perform hypothetical experiments by implementing management scenarios in simulations of the SES;

- To interpret the outcome of the hypothetical experiments. Subsequently, to gain insight about (1) the above-said recognition scheme and its potential for promotion of environmentally responsible behavior, and (2) the state of health of the ecological resource in response to the social dynamics that emerge from the recognition scheme.

## 2 Methods

To answer the questions raised in introduction, we take a modelling and scenario analysis approach. Given the complexity of the subject of our study and considering the precautionary principle (Kriebel et al., 2001), it is desirable to avoid the risk of irreversible damage that is associated with a trial-and-error learning approach. Therefore, we opted for modelling, which allows to design and repeatedly perform experiments without the risk of adverse irreversible effects. In other words, a model can serve as a virtual laboratory to safely run hypothetical experiments. SES governance literature highlights the use of conceptual models for gaining insight about complex SES problems (Janssen and Ostrom, 2006b).

The environment in which we run hypothetical experiments is a conceptual model, which is constructed by coupling two previously developed base models. In this section we introduce the base models, the mechanism for coupling the base models, and the scenarios for testing the coupled model. Appendix A presents further details of the models according to the ODD+D protocol, which is an extension of the ODD protocol particularly adapted for describing human decisions in ABMs (Müller et



al., 2013). The ODD (Overview, Design concepts, Details) protocol is a standard for communication of information about ABMs (Grimm et al., 2006, 2010, 2020).

## 2.1 Ecological model

This model (Harati et al., 2020, 2021b) simulates the spread of a forest insect infestation. The model is built based on observed

data of the Mountain Pine Beetle outbreaks in the province of British Columbia in western Canada. In this model, the study area is represented as a grid with each cell containing geospatial information as well as binary data on presence or absence of infestation. For each cell, based on its geographic variables and distance weighted sums of infestations in its neighborhood, the model predicts the state of infestation in the next time step. This model uses logistic regression because this machine learning algorithm is fast and suitable for applications with binary response variable. Model calibration is dependent on the

study area. In this work, the study area is a division of Kamloops timber supply area in British Columbia, Canada, with extents from 120°19'59"W 50°45'22"N to 119°6'0"W 51°32'40"N. Figure 1 shows the study area.

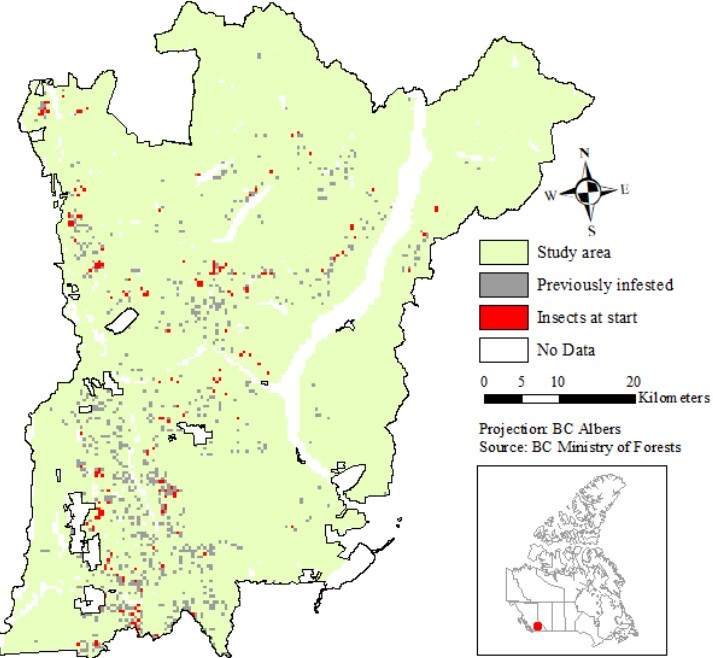

**Figure 1: Study area of forest insect infestations simulation in British Columbia, Canada. Locations of active infestations at the beginning of the simulation period are marked as 'Insects at start'. Locations that were infested before the start of simulations are**
**marked as 'Previously infested'.**

## 2.2 Social model

This conceptual model simulates interactions between individuals in a problem known as Principal-Agent and provides insight on the emergence of social norms. In the Principal-Agent problem the goal of one or more Principals is that one or more Agents perform a certain behavior (Braun and Guston, 2003). In our social model, the desire for good reputation is used as a motivation





for Agents to perform the behavior that the Principal desires. In this conceptual setting, the Principal grants recognition to Agents who act as the Principal requests. The model runs in discrete time steps. In each time step the Principal chooses the cost of its request from the Agents. Specifically, the Principal makes a binary choice between a costly request and a no-cost request. The Agents, on the other hand, compare the cost of the requested action with the benefit of receiving recognition, and decide whether or not to cooperate with the Principal. Agents assess the benefit of recognition as a function of (1) the existing

awareness in their society of such recognition, and (2) expectation of uniqueness among their peers if they obtain recognition in the next time step. This design is inspired by existing literature, which underscores the importance of personal motivations such as good reputation in voluntary action (Omoto and Snyder, 1995; Stern et al., 1993) and the importance of visibility of one's actions in one's behavioral choices in society (Mosler, 1993; Nyborg et al., 2016).

The model uses a Temporal Difference Reinforcement Learning algorithm named Double Expected SARSA to develop a

decision making guideline for the Principal. Input to this decision guideline is the behavior of Agents, and output is the binary choice of action to request, i.e. costly or no-cost. The objective of the algorithm is that by the end of simulations, an acceptable proportion of Agents cooperate with the Principal in its costly request. Such acceptable proportion is assumed 0.5 in this study in order to represent the choice of the majority. In Reinforcement Learning literature, the decision making guideline developed and used through the algorithm is called *policy*. In the study presenting the social model (Harati et al., 2021a) three levels of

low, medium and high were assumed for the cost of the Principal's costly request. Note that each of these levels is contrasted with a no-cost action in the binary decisions of the Principal. For each of the above three levels the model was run and corresponding *policy* was calculated.

## 2.3 Coupled social-ecological model

In this work we added a coupling mechanism to the above two models and built a conceptual social-ecological model to

simulate a hypothetical setting. In this setting, in the ecological model, the forest is attacked by MPB. These attacks spread in each iteration of the model, which represents one year. In the conceptual social model, there is a governing agent whose goal is to stop the spread of infestations, and several user agents, which represent logging companies. The governing agent intends to create a buffer zone between insects and healthy forest by cutting the forest in vicinity of newly observed infestations. This intervention measure is difficult and costly, and the governing agent intends to use the social model's recognition mechanism

(Harati et al., 2021a) and encourage user agents to voluntarily perform the intervention action. Each user agent of the social model is assigned to a part of the simulated forest of the ecological model. Decisions of the user agents are implemented in their respective forest areas in simulations.

In each iteration of the model, the governing agent decides whether or not to ask the user agents to do its intended action, that is, cutting trees around newly observed infestations. In case the governing agent asks for that action, it grants recognition to

cooperating users. Such recognition is a responsible user label. In case the governing agent does not ask for cooperation in the intervention measure, the governing agent grants recognition labels to all user agents. On the other hand, each user agent, when requested to perform the intervention measure in exchange for the recognition label, makes a decision by comparing the cost





of the requested action versus the benefit of recognition. The cost of action is assessed based on the forest area to be cut by the user, and the value of the recognition label is assessed based on the society's knowledge of that label as well as the expectation

of being unique in having that label in the next time step.

In the making of the model, we noticed that before the emergence of cooperative behavior in the social model, the forest would be largely infested in the ecological model. While this observation was insightful, it also showed that our simulations were reduced to runs of the ecological model alone. In other words, in absence of intervention from the social model, the ecological model would simulate spread of infestations as if the two models are not connected. Because our interest was to study the

complexities that arise from the interactions between the two models, we added a new feature in our setup. We defined preparation steps, during which the social model runs alone before being coupled to the ecological model. The preparation steps can be considered as awareness campaigns in the society. Through these steps, the governing agent and user agents interact according to the rules of the social model (Harati et al., 2021a). Consequently, user agents become familiar with the recognition mechanism and the responsible user label.

To implement the idea of coupling the two models, we defined a mechanism which we named flip-flop. Recall that the ecological and social models run in R and Java environments, respectively. To each model we allocated a directory, named Inbox, in computer hard disk. Each iteration of the coupled model begins with the ecological model simulating spread of infestations and while the social model waits in a loop, constantly checking its inbox for a new message. Based on newly observable infestations in the simulation, the ecological model produces a report indicating the size of intervention buffer zone

in each user agent's designated forest area. This report, which is a text file, is then copied in the Inbox directory of the social model. At this point, the social model notices the new file in the Inbox folder, exits the waiting loop, reads the file and proceeds to simulate interactions between governing and user agents. Contents of the received file are needed in the cost-benefit analyses done by user agents. The social model, in turn, produces a report indicating the user agents who intend to cut trees in the intervention buffer zone of their respective forest areas. This report, also a text file, is copied in the Inbox folder of the

ecological model, which by this time has been waiting in a loop and constantly checking its Inbox for a new message. The ecological model then exits its waiting loop, reads the file sent by the social model, and accordingly changes the landscspe of simulations by eliminating forest cover in buffer zones in areas associated with indicated user agents. Such updated landscape will be the basis for simulations of the next time step. Figure 2 depicts the concept of the flip-flop mechanism for coupling the two models, and Fig. 3 shows a more detailed view of the coupled model.

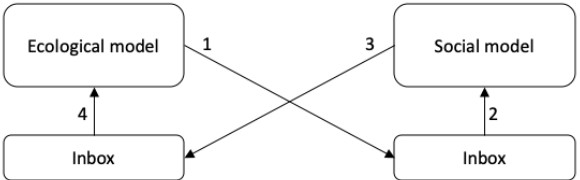


**Figure 2: The flip-flop mechanism. Each of the two models requires input from the other model. The models communicate with each other via inbox directories. Arrows show direction of data transfer. Numbers beside arrows show the order of operations.**



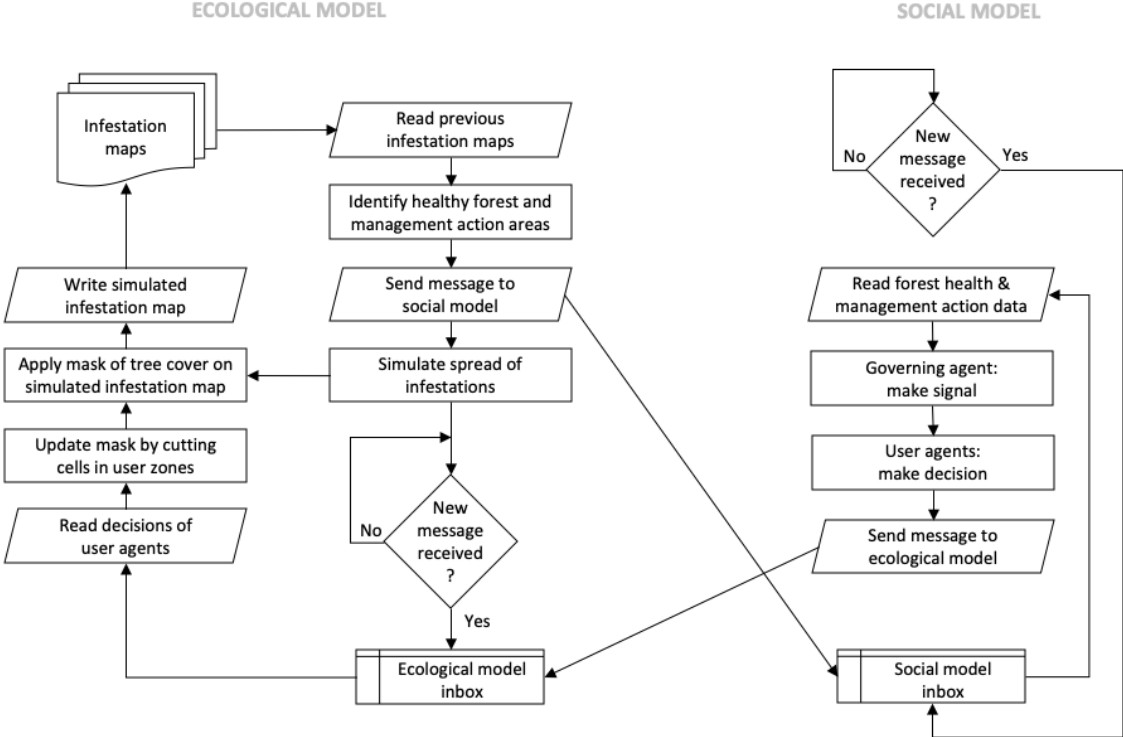

**Figure 3: The coupled model. In each time step, the ecological model begins with reading previous infestation maps and ends with**
**writing a newly simulated infestation map; the social model begins with waiting for a message from the ecological model and ends**
**with sending and message to the ecological model.**

### 2.4 Simulation scenarios

In order to gain insight about emergence of environmentally responsible behavior in the setting described above, we ran several rounds of simulations with different scenarios. We defined scenarios so that the comparison of their results provides useful
information about the subject of study. Below are descriptions of these scenarios:

1. The simplest scenario is Business As Usual (BAU), in which there is no intervention from the governing agent to control the disturbance. In each time step, user agents harvest a proportion of their allocated zones. That proportion is the business-as-usual harvest ratio, which is a model parameter. The spatial ecological model iteratively simulates the spread of infestations, noting that harvested grid cells cannot become infested anymore. In this scenario, the
governing agent's RL algorithm is not engaged. This scenario indicates a case where there is no government intervention to control the insect disturbance, or a case where user agents never cooperate with the governing agent. Hence, this scenario serves as a baseline for comparison with main simulations.

2. Another baseline in our study is a scenario in which all user agents always cooperate fully with the governing agent. This scenario, which we named Enforce, provides a best case for the social component of our study. The Enforce





scenario shows the effectiveness of the management plan in the control of the disturbance. In this scenario the governing agent's RL algorithm is not engaged.

3-      Our main scenarios, which we named Suggest, are those in which the governing agent is active and uses its RL algorithm. In Suggest scenarios, the governing agent suggests that if user agents cooperate with it in the management action then they will receive '*responsible user*' labels. User agents then analyze the governing agent's suggestion and

make their decisions. In terms of cooperation of user agents with the governing agent, the Suggest scenarios are between BAU and Enforce. The neighborhood of management action is a Moore neighborhood of the newly visible infestations. The size of this neighborhood is a model parameter. We ran simulations with a neighborhood of size 4 grid cells. We also defined preparation runs, in which the ecological model is not engaged. Instead, agents in the social model interact with each other, which results in increased visibility and value of the '*responsible user*' label.

Thereupon, the following three scenarios were defined:

o    Suggest scenario with 0 prepation time steps

o    Suggest scenario with 10 prepation time steps

o    Suggest scenario with 20 prepation time steps

4-      Corresponding to each Suggest scenario, we defined another baseline, in which the governing agent behaves randomly

instead of using its RL algorithm. In these scenarios, which we named Random, user agents analyze and respond to the governing agent's signals, as in the Suggest scenarios. The calculation of the state of health of the resource and the costs of management action in user agent zones are performed similar to the Suggest scenarios. The only difference between Suggest and Random scenarios is in the decision making mechanism of the governing agent. In this sense, by showing what could be achieved with a naïve model, Random scenarios serve as a baseline to indicate the power

of the sophisticated RL algorithm of the governing agent. Thereupon, the following three scenarios were defined:

o    Random scenario with 0 prepation time steps

o    Random scenario with 10 prepation time steps

o    Random scenario with 20 prepation time steps

**3 Results**

Figure 4 shows the mean maps of remaining infestations in the simulated scenarios at the final time step. It can be seen that, without preparation, the Suggest and Random scenarios are similar to BAU. On the other hand, with addition of preparation steps, less infestation remains in the study area. The figure also shows that in Suggest scenarios less infestation remains in comparison with Random scenarios.



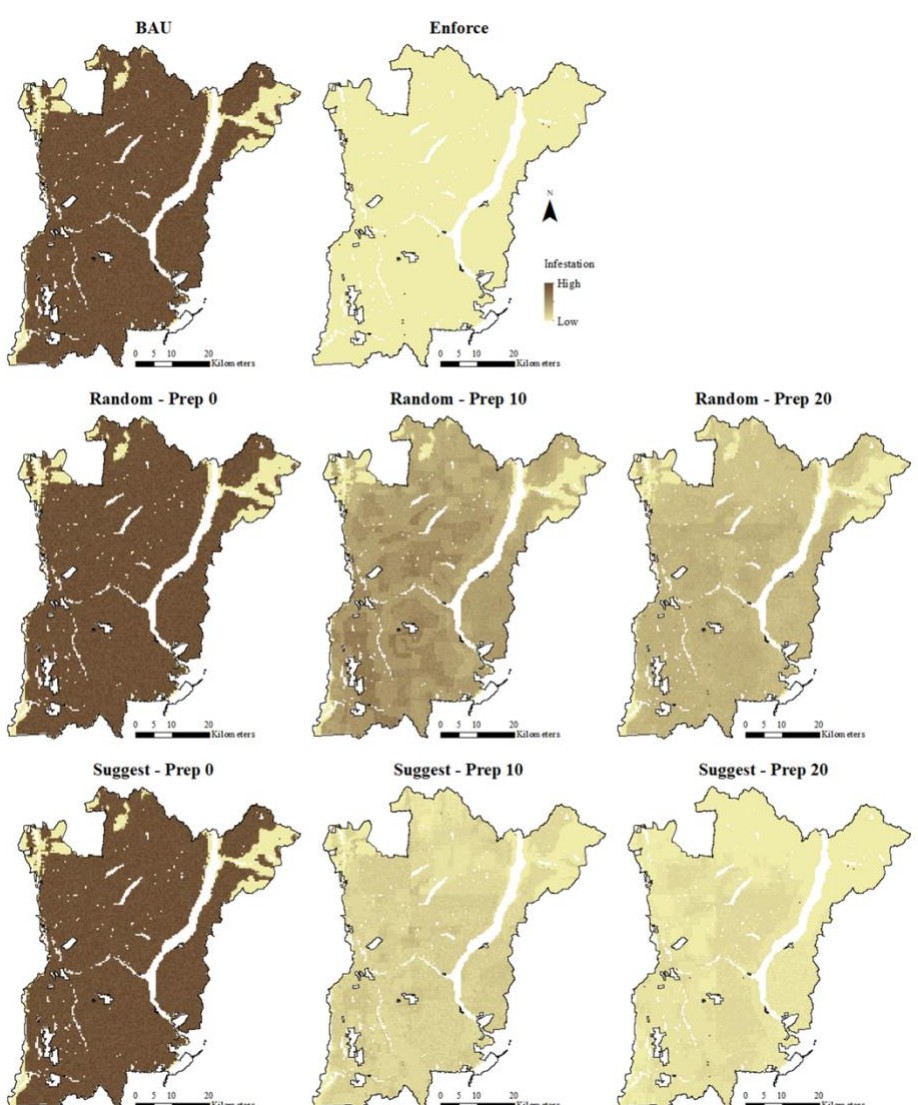

**Figure 4: Maps of mean remaining infestation after the final time step of simulations. The Enforce scenario was run once, and all other scenarios were run 50 times. For each scenario, 'High' infestation in a cell means the presence of infestation in the cell in all runs of that scenario; 'Low' infestation means the absence of infestation in the cell in all runs of that scenario.**

Figure 5 shows the mean ratio of cooperation of user agents with the governing agent over time steps of Suggest and Random scenarios. Without preparation of the user agents, both Suggest and Random scenarios end with nearly no cooperation at all. Therefore, in these cases no management action is done to control the infestations, which explains why the maps of no-preparation scenarios are similar to the map of BAU. With preparation, cooperation ratio increases in both Suggest and Random scenarios, with Suggest scenarios showing higher cooperation than Random. Nonlinear behavior is observed in the curves of





Suggest and Random scenarios with 10 steps of preparation, which shows sudden emergence of cooperation with the governing agent.

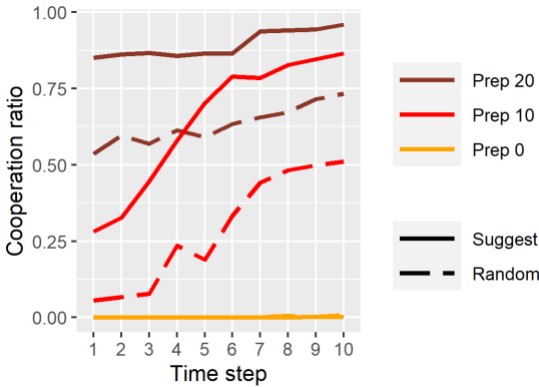


**Figure 5: Mean cooperation ratio over time for Suggest and Random scenarios with 0, 10 and 20 preparation steps. Each scenario was run 50 times.**

Figures 5 and 6 show the mean proportions of study area that are covered by healthy and infested forest, as well as the area that is cut, in each time step. The baseline scenarios BAU and Enforce, which are shown in Fig. 6, indicate the maximum

amount of forest that can be saved from infestation if the management action is successfully implemented. The Random and Suggest scenarios, shown in Fig. 7, demonstrate interim situations where the management action is partly implemented. The plots of Random and Suggest scenarios also show the results of adding preparation steps in the simulations.

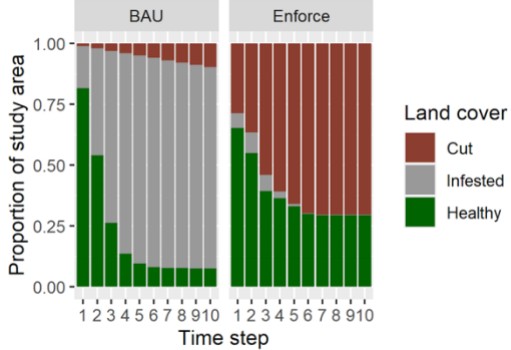

**Figure 6: Proportions of healthy, infested and cut areas in BAU and Enforce baseline scenarios.**





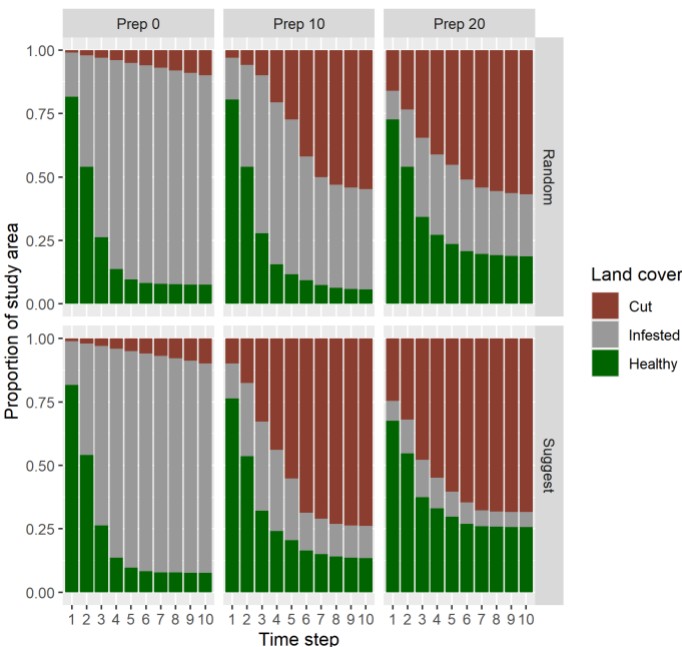


**Figure 7: Proportions of healthy, infested and cut areas in Random baseline and Suggest scenarios. Each plot represents 50 runs.**

Table 1 gives a quantitative summary of the proportions of healthy, infested and cut area at the end of the simulations. Note that the Enforce scenario was run only once, because it involves no stochasticity in decisions of agents. This is why there is no deviation in the results of this scenario. The mean values in this table correspond to the final time step in the plots of Figs. 5

and 6. The standard deviations reveal several things about variations of the results. Firstly, variations are minimal in the BAU, as well as in Random and Suggest scenarios with no preparation. These are the scenarios in which user agents rarely cooperate with the governing agent, and therefore, the management action is not implemented. Variations in results increase substantially when preparation steps are included in simulations. Secondly, variations in the proportion of healthy area are smaller than variations in proportions of infested and cut areas. Thirdly, variations in Suggest scenarios are smaller than variations in

Random scenarios. This is particularly evident in scenarios with 20 preparation steps. This table shows that, in comparison with the Random baseline scenarios, the Suggest scenarios result in a higher proportion of healthy forest at the end of the simulations, with smaller variations in results.

**Table 1: Mean and standard deviation of the final proportions of healthy, infested and cut areas in simulations. Each scenario was run 50 times, except for Enforce, which run once.**





| Scenario | Healthy proportion | | Infested proportion | | Cut proportion | |
|---|---|---|---|---|---|---|
| | Mean | S.D. | Mean | S.D. | Mean | S.D. |
| BAU | 0.076 | 0.001 | 0.827 | 0.001 | 0.097 | 0 |
| Enforce | 0.295 | 0 | 0.001 | 0 | 0.704 | 0 |
| Random-Prep 0 | 0.075 | 0.003 | 0.825 | 0.008 | 0.100 | 0.010 |
| Random-Prep 10 | 0.057 | 0.037 | 0.395 | 0.292 | 0.547 | 0.297 |
| Random-Prep 20 | 0.187 | 0.110 | 0.245 | 0.321 | 0.568 | 0.243 |
| Suggest-Prep 0 | 0.076 | 0.002 | 0.826 | 0.002 | 0.098 | 0.004 |
| Suggest-Prep 10 | 0.135 | 0.084 | 0.126 | 0.249 | 0.739 | 0.239 |
| Suggest-Prep 20 | 0.257 | 0.089 | 0.059 | 0.166 | 0.683 | 0.110 |


It can be seen in Figs. 6 and 7 as well as Table 1 that results of the no-preparation scenarios are similar to BAU. On the other hand, as shown in Fig. 5, in runs with more preparation steps, the percentage of cooperation of user agents with the governing agent increases. Such increase is larger when the governing agent's decisions are made by the RL algorithm, i.e., in Suggest scenarios. Comparing the zero-preparation and 20 step-preparation scenarios in Figs. 6 and 7, it is evident that the 20 time

steps of preparation lead to an increase in the remaining healthy forest, even when the governing agent's decisions are random. Considering all cases with preparation of user agents, it can be observed that more healthy cells are saved in Suggest scenarios, i.e. with RL decision making algorithm for the governing agent than in Random scenarios, i.e. with random decisions. The case of 10-step preparation with random decisions of the governing agent is particularly noteworthy. In this scenario, first the infestation spreads to large areas, and then the management action begins, which involves cutting large areas around the newly

observed infestations. Consequently, the remaining healthy forest in this scenario is even slightly smaller than BAU.

Tables 2 and 3 summarize non-parametric Wilcoxon signed rank tests that were performed to statistically analyze simulation results. Scenarios were compared in terms of proportions of remaining healthy forest after the final time step of their respective simulations. As can be seen in Table 2, the proportions of remaining healthy forest in scenarios Suggest-Prep10, Suggest-Prep20 and Random-Prep20 are greater than BAU. Likewise, in comparison with Enforce, all scenarios lead to significantly

less proportions of remaining healthy forest, except for Suggest-Prep20. In other words, the result of Suggest-Prep20 is very similar to our best cooperation case baseline. Table 3 shows the comparison of Random and Suggest scenarios. It is seen that without preparation steps, the results of Random and Suggest scenarios are not significantly different. On the other hand, in scenarios with preparation steps, the proportion of remaining healthy forest is greater in Suggest scenarios than in Random scenarios.

**Table 2: Wilcoxon signed rank test statistic and p-value for comparison of remaining healthy forest proportions of scenarios with BAU and with Enforce (n=50).**





| | Scenario | Null hypothesis | Alternative hypothesis | Statistic | p-value |
|---|---|---|---|---|---|
| Comparison with BAU | Random-Prep 0 | | | 638.0 | 0.50 |
| | Random-Prep 10 | | | 295.0 | 0.99 |
| | Random-Prep 20 | Scenario = BAU | Scenario > BAU | 1133.0 | $8.8 \times 10^{-07}$ |
| | Suggest-Prep 0 | | | 669.5 | 0.38 |
| | Suggest-Prep 10 | | | 1064.0 | $1.9 \times 10^{-05}$ |
| | Suggest-Prep 20 | | | 1244.0 | $2.4 \times 10^{-09}$ |
| Comparison with Enforce | Random-Prep 0 | | | 0 | $3.8 \times 10^{-10}$ |
| | Random-Prep 10 | | | 0 | $3.8 \times 10^{-10}$ |
| | Random-Prep 20 | Scenario = Enforce | Scenario < Enforce | 300 | $4.8 \times 10^{-04}$ |
| | Suggest-Prep 0 | | | 0 | $3.8 \times 10^{-10}$ |
| | Suggest-Prep 10 | | | 15 | $9.5 \times 10^{-10}$ |
| | Suggest-Prep 20 | | | 903 | 0.99 |

**Table 3: Wilcoxon signed rank test statistic and p-value for comparison of remaining healthy forest proportions between Random and Suggest scenarios (n=50).**

| Null hypothesis | Alternative hypothesis | Statistic | p-value |
|---|---|---|---|
| Random-Prep0 = Suggest-Prep0 | Random-Prep0 < Suggest-Prep0 | 555.5 | 0.29 |
| Random-Prep10 = Suggest-Prep10 | Random-Prep10 < Suggest-Prep10 | 140.0 | $8.0 \times 10^{-07}$ |
| Random-Prep20 = Suggest-Prep20 | Random-Prep20 < Suggest-Prep20 | 109.0 | $5.7 \times 10^{-03}$ |


## 4 Discussion

### 4.1 Insights about the case of study

Our simulation results reveal several points that deserve further attention and discussion. One such point is the importance of the first time step in the result of the simulations. In the comparison of the scenarios in Fig. 7, those which have a smaller
proportion of infested cells at the end of the first time step have a larger proportion of healthy forest remaining at the end of the final time step. Future infestations as well as the size of the areas to cut thus depend on infestations in the first time step. Therefore, the more newly infested trees are cut in the first time step, the more healthy forest remains in future time steps. As an implication of the importance of success at the first step, and expanding this discussion beyond the scope of the present study, if the government has access to limited financial resources to provide incentives for users, then our insight suggests that
focusing the incentives at the beginning of intervention may contribute substantially to success in the end. The study of such





cases and building hybrid models for them, where the government uses both incentives and recognition, can be an area for future research.

Our results in Fig. 7 show that forest cover is eliminated not only by infestations, but also by the considered management action, which is to cut trees in order to stop the progress of infestations. Depending on the scenario, the proportion of the forest

that is cut may even be larger than the proportion that is infested. Particularly, when infestations spread in a large area and user agents decide to cooperate with the governing agent, user agents will cut a large zone in the forest and a small proportion of healthy forest will remain. Likewise, in the Enforce scenario, control of the epidemic is achieved at the cost of cutting a large area of the forest, but in this case the proportion of remaining healthy forest is larger, and the participation of user agents is enforced, not voluntary.

An assumption in the simulations of this study is that for the user agents the perceived cost of cooperation with the governing agent is high. This cost appears as a threshold in the decision process of user agents, and it is compared against the calculated utility of cooperation with the governing agent. In the study that presented and described our social model (Harati et al., 2021a), user agents' perceived cost of cooperation with the governing agent is defined in three levels of low, medium and high, where high cost of an action means low motivation to perform that action. Therefore, the present study involves a challenge for the

social model, because our user agents are defined to be hesitant towards cooperation with the governing agent. In such challenging problems with high cost of the desired behavior, the governing agent may succeed in promoting the desired behavior with appropriate decision making algorithms but not with random decisions (Harati et al., 2021a). The technique that we added in the present study was the inclusion of additional preparatory steps. We found that preparatory steps lead to the emergence of the desired behavior among user agents, even if the governing agent's decisions are made randomly. This shows

that recognizing responsible users and introducing them to the society is a powerful mechanism with the potential to create new behavior norms.

The governing agent's RL algorithm uses a policy for making decisions, and updates that policy based on the rewards it receives as a result of its decisions. The governing agent is not aware of the decision thresholds of user agents. In the study that defined the social model (Harati et al., 2021a), in each of the cases with low, medium and high user agent thresholds, the

governing agent's RL algorithm was trained and calibrated according to the behavior of the user agents. Therefore, corresponding to each case of user agent thresholds a policy was calculated for the governing agent. In our present study, still assuming that the governing agent does not know if the decision thresholds of user agents are low, medium or high, we parameterized the governing agent with the policy calculated in training the RL algorithm with medium threshold user agents in the previous work (Harati et al., 2021a).

Another interesting matter about the simulations is that they show that, in all scenarios, infestations spread rapidly at first and slow down later, such that by the final time steps little or no change is noticeable in the proportion of the study area that is infested. As the spread of infestations stops, there will be no areas to cut around observed infestations, and the composition of the study area does not change anymore. It is noteworthy that in BAU the largest spread of infestations happens in the first





three time steps. Therefore, in the Suggest scenarios, if no management action is taken in these initial time steps, then a large
part of the forest is destroyed by infestations.

A feature of our simulations was the definition of the management action. Cutting neighborhoods of infestations is only one
of the possible actions to control infestations (Maclauchlan and Brooks, 1994). Our inspiration for choosing this action came
from a previous work, in which we performed spatial analyses of spread of MPB infestations and validation tests on our land
change model (Harati et al., 2021b). In that work, we noted that most of the new MPB infestations occur in the vicinity of
previous locations of attacked trees. Therefore, in the present study we defined neighborhoods of management action around
newly observed infestations. The action was to cut the cells in these neighborhoods. Our choice of neighborhood size was also
inspired by distance analysis in that previous work (Harati et al., 2021b) as well as the consideration that the governing agent
cannot detect new infestations in the first time step after their occurrence. This is due to the fact that infested trees do not
change color in the first year after attack. It follows that infestations spread further in a larger neighborhood before the
governing agent realizes their previous locations.

Through the course of MPB infestations, BC eventually adopted the policy of increasing the allowable annual cut, first in order
to suppress the infestations, and later to facilitate salvage harvest in infested areas (Forest Practices Board, 2007, 2009). Such
increase was smaller than the simulations of the present study. In severely infested areas of the province, the allowable annual
cut was increased by 80 percent of the pre-infestation levels (idem), which was less than one third of a percent of the forest
area (BC Ministry of Forests, 2003). Therefore, at the largest increase of the allowable annual cut, still less than one percent
of the forest was harvested per year.

Cutting trees in large scale can be a practical challenge. According to an analysis of data of year 2008, with that year's rate of
harvesting, it would take 22 years to cut the pine trees that were killed by infestations up to 2008 (Forest Practices Board,
2009). Moreover, although the government wanted the added harvesting to be concentrated in severely infested areas to control
the pest, the forest industry preferred to harvest from other locations and especially from forest stands with mixed species
(Forest Practices Board, 2007). Furthermore, the decision to increase the allowable annual cut raised concerns about possible
ecological impacts (Forest Practices Board, 2007, 2009), which is beyond the scope of the present work and can be studied in
future research.

### 4.2 Insights about governance of SES

In addition to the above-mentioned points about the particular case of simulations, in this study we gained insight into the more
general problem of sustainable management of SESs by engaging users. Our most important finding in this study is that it is
possible to create a strong motivation for effective action towards protection of natural resources in a SES by encouraging
users – without financial incentives, enforcement or punishment. The latter is of particular importance because the role of
punishment as a basis for the formation of norms of environmentally responsible behavior has been emphasized in the literature
of SES (Farrow et al., 2017) and social sciences (Axelrod, 1986). Our results show that even without punishment, recognition
of responsible behavior through the mechanism of our model can create a force towards emergence of a norm of responsible





behavior. Our approach, which is only one of the possible approaches to the problem of collective action, is based on the theory of normative conduct (Cialdini et al., 1990). Moreover, from model results presented in Fig. 7 we gain the insight that even with existence of the potential for action towards protection of a natural resource, uncalculated governance decisions about

using that potential may cause adverse effects on the resource. The simulations show an example of this type. The difference between model results with and without an intelligent algorithm highlights the importance of well-thought-through decision making in governance of SES.

Another general insight that we gained from the simulations pertains to the temporal differences between social and ecological processes, which add to the complexity of a SES. For example, from our simulations we noted that formation of an

environmentally responsible behavior norm takes time. We also noted, throughout the simulations produced, that the largest damage made by the ecological disturbance occurred in the beginning time steps of the study. Therefore, if the efforts to promote a new norm of responsible behavior begin at the onset of the ecological disturbance, then by the time the responsible behavior emerges in the society much of the resource is already lost. This means that it is important to prepare the society and promote environmentally responsible behavior before there is a need for action to protect the natural resource.

**4.3 Challenges and perspectives for future work**

Our model is defined in a specific scope. This scope might as well be considered a limitation for the model. That is, our model does not account for the effect of other processes than what we included in it. Future works may use other social models instead of ours and insert them in our SES model. Potential research efforts may as well couple other ecological models to the social model. In these cases, the coupling mechanism of our model may be modified according to the needs of other applications and

assumptions. A challenge that can be addressed in future works is the addition of a third submodel in the coupled model to account for economic complexity. In the example of this study, total harvest from the forest comprises market supply, which influences market price and sales quantity, which in turn influence users' profit and hence individual decisions on harvest in the next time step. Another matter that may be considered in future works is the interactions among user agents. In the present study our goal was to gain useful insight for managerial and government decision making. Therefore, our RL decision making

algorithm was placed in the governing agent. Future works can equip user agents with more sophisticated decision algorithms.

**5 Conclusion**

We connected a social model and an ecological model, which were developed independently, through a coupling mechanism to build a conceptual social-ecological model. Using our model we carried out tests that allowed us to perform 'what-if' analyses with several scenarios of SES management. Our simulations showed that in a society where individuals or companies

(i.e. "agents") care about their reputation, it is possible to promote environmentally responsible behavior through an encouragement mechanism, without use of force, without use of financial incentives, and only by recognition of responsible individuals in the society. In the management of a SES under disturbance, it is important to note that before the emergence of



environmentally responsible behavior, the disturbance may damage the ecological system, as demonstrated in our zero-preparation scenario simulations. It is therefore important to prepare the society in advance for engagement in environmental

protection and ecological conservation action. We used a conceptual model as a virtual laboratory for performing hypothetical experiments, and by comparing the outputs of those experiments we gained insight about a complex system.

**Funding.** This research was partially funded by the Natural Sciences and Engineering Research Council (NSERC) of Canada through the Discovery Grant number RGPIN/05396-2016 awarded to L.P., R.M.-H. received financial support from the

European Union's Seventh Framework Programme through NEWFOREST programme number PIRSES-GA-2013-612645.

**Code/data availability.** Model code files are available at Zenodo repository (https://doi.org/10.5281/zenodo.11245520) under MIT license. Datasets of model input and output are available at OSF repository (https://doi.org/10.17605/OSF.IO/URJQ8). Further information about the model is presented below.

Model name: flipflopSEM (FlipFlop: a Social Ecological Model)

Developers: Saeed Harati-Asl, Liliana Perez, Roberto Molowny-Horas

Languages: Java, R

**Author contribution.** S.H.-A. developed model concept and code under supervision by L.P. and R.M.-H. S.H.-A curated model data and ran the model. All authors contributed to writing and the preparation of the manuscript.

**Competing interests.** The authors declare no competing nor conflicting interests.

**Acknowledgements.** We are thankful to Université de Montréal's International Affairs Office (IAO) for their financial support through the International Partnership Development program, which allowed the collaboration between researchers from Université de Montréal and CREAF.

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

**Appendix A – Model description using ODD+D protocol**

**A.1 Overview**

**A.1.1 Purpose**

The overall goal of this study is to gain insight about the possibility of emergence of environmentally responsible behavior in a SES in the absence of altruism, obligation, and financial incentives. To that end, this model has been built with the purpose of simulating a mechanism of recognition of environmentally responsible behavior in a setting of forest disturbance. We use the model to learn about the complexities of a SES in which users of a forest resource are requested to participate in a costly action to protect the resource from a disturbance. Particularly, we intend to better understand the potential of the desire for good reputation in promotion of environmentally responsible behavior, and its implications in management and policy making. The model has been designed for scientists, policy and decision makers, and experts with an interest in developing decision support systems.



#### A.1.2 Entities, state variables and scales

The model consists of a social component, which is an ABM, and an ecological component, which is a spatial ecological
model. The social ABM includes a governing agent, several user agents and an auxiliary agent called registrar, which has been
defined for a better understanding of the model. The model includes a spatial component, which represents a forest resource
attacked by insect infestations. The spatial ecological model's units are grid cells.

In each time step the governing agent is in a state, takes and action, and receives a reward. The governing agent's state is a 2-
dimensional variable which cumulatively summarizes past interventions and their results. The two components of this state
variable are calculated based on actions and rewards. The governing agent's reward is the cooperation by the user agents in
the management action to save the forest. The governing agent's action is to generate a binary *signal* for communication to
user agents. A signal of 0 is a request for a no-cost action, and a signal of 1 is a request for a costly action. To produce *signal*,
it uses a *policy*, which recommends an action for each state. To update its *policy*, it uses its memory of past states, actions and
rewards. For that purpose, the governing agent has two arrays of time-discounted scores calculated for each *(state,action)* pair
(Harati et al., 2021a). Each user agent is allocated a forest zone where the user agent harvests. These zones are created by
dividing the map of the study area into equal squares. Each square contains cells that are in the study area and cell that are not.
Therefore, user zones include various numbers of cells from the study area. Moreover, each user agent is characterized by a
constant decision *threshold* that it uses in a cost-benefit analysis to make a binary *decision* in response to the governing agent's
*signal*. Decisions of 0 and 1 mean rejection and acceptance of the governing agent's request, respectively. The registrar is
characterized by two variables, *nSum* and *nLast*, which are non-negative integers. *nSum* is the total number of times user agents
decided to cooperate with the governing agent. *nLast* is the number of user agents who cooperated with the governing agent
in the last time the latter requested a costly action. In the spatial component of the model, each cell is identified by geographical
data fields including coordinates, elevation, aspect, slope and ruggedness. Cells are marked with presence or absence of
infestations. Cells are also marked with a mask layer that indicates presence or absence of trees. The simulation area is divided
into zones and each zone is allocated to a user agent. Table A.1 lists the model entities and their state variables.

**Table A.1: An example for accurate data representation & universal readability of figures.**

| Entity | State variable | Description |
|---|---|---|
| Registrar | nLast | Last known number of user agents cooperating with the governing agent |
|  | nSum | Cumulative number of user agents cooperating with the governing agent |
| Governing agent | state | 2-dimensional summary of past interventions and results |
|  | signal | Binary request (easy task or hard task) |
| User agent | decision | Binary response (refuse or accept) |
| Cell | infestation | Binary indicator of presence or absence of infestations |
|  | mask | Binary indicator of presence or absence of host trees |



The social ABM is influenced by an exogenous driver: spread of insect infestations in the forest. This driver is simulated in the spatial ecological model, which is coupled with the ABM. The spatial ecological model is a GIS model that simulates land change. The land change process of this study is the infestation of forests of British Columbia (BC) in western Canada by the

Mountain Pine Beetle (MPB). The area where infestations are modeled is a sub-division of the Kamloops Timber Supply Area (TSA). The extents of this area are from 120°19'59"W 50°45'22"N to 119°6'0"W 51°32'40"N. In the spatial ecological model, one grid cell represents an area of 400m x 400m.

A time step in the coupled social-ecological model represents one year and the simulations ran for 10 years. In addition, separate sets of simulations were run with the social model alone, in which time step was arbitrarily defined as one month.

These simulations, which prepared the agents for later runs of the coupled social-ecological model, were performed with 10 and 20 additional steps.

### A.1.3 Process overview and scheduling

In the coupled social-ecological model, in each time step, infestations spread from infested cells in the previous time step. This spread is simulated by the spatial ecological model. Newly infested grid cells remain invisible to the governing agent for one

time step after infestation and become visible in the next time step. These grid cells act as sources of spread of infestations while they are invisible to the governing agent.

The social model's governing agent analyzes the last visible spread of infestations and calculates the cost of management action to stop the spread of infestations in each user zone. The management action is to cut a neighborhood area of the forest surrounding last visible infestations. Accordingly, the cost of action is defined proportional to the size of the said neighborhood.

Then the governing agent sends a binary signal to the user agents. A signal of 1 means the governing agent requests the user agents to participate in the management action in their allocated forest zone voluntarily and at their own cost, in exchange for a 'responsible user' label. The 'responsible user' label only shows recognition of the user agents who cooperate with the governing agent, and has no monetary value. A signal of 0 means the governing agent requests the user agents to do an easy task with no cost for the user agents and no benefit for the governing agent, in exchange for the 'responsible user' label. There

is no difference between labels given when the governing agent's signal is 0 or 1. The governing agent uses a Reinforcement Learning (RL) algorithm in order to produce its signal, considering the past states, actions, and rewards (Harati et al., 2021a). The reward for the governing agent is the cooperation by the user agents in the costly management action and thus saving the forest from the infestations.

Each user agent considers the governing agent's signal and produces a binary decision in response, which indicates whether

or not the user agent accepts the governing agent's request in exchange for the 'responsible user' label. When the governing agent's signal is 0, the requested action is of no cost and all user agents accept the governing agent's request and all user agents receive the 'responsible user' label. When the governing agent's signal is 1, each user agent makes a decision with a cost-benefit analysis, taking into account the governing agent's calculated cost of action, history of the 'responsible user' label, the uniqueness and visibility that they gain if they get the label, and the revenue from sales. Each user agent considers obtaining





the label as an opportunity to be unique in having a recognition that some user agents do not. Such uniqueness is assessed based on the response of user agents to the governing agent in the last known interaction. Each user agent also assumes that the acknowledgement of the label in its group depends on how much the group knows the label, which in turn depends on the cumulative number of times the label has been seen in the group. Thereupon, user agents consider a visibility score for the label.

In order to help understand interactions in the model, another agent, the registrar, is defined. The registrar observes and registers actions of the governing agent and the user agents in each time step from the beginning of an episode of simulations. The other agents refer to the registrar in their decision making process.

Once the users make their decisions, a message is sent from the social ABM to the spatial ecological model, and modifications are correspondingly made in the forest map. These modifications include removing trees for annual harvest or management

action. Specifically, each user's zone is subject to annual harvest unless that user cooperates with the governing agent when the signal is 1. In this case, that is, if the signal is 1 and the user agents cooperates with the governing agent, the neighborhood indicated for management action in the user's zone is cut. The modified landscape map is used by the spatial ecological model in the next time step.

## A.2 Design concepts

### A.2.1 Theoretical and empirical background

The core idea of the social ABM is the promotion of responsible behavior using individuals' desire for respect. The theoretical basis for this idea notes that sustainability issues are problems of collective action (Ostrom, 1990), that an individual's behavior is influenced by the observation of behavior of others in the society, or descriptive norms, as stated in the theory of normative conduct (Cialdini et al., 1990), and that people care about their reputation in the society (Anderson et al., 2015; Lazaric et al.,

2020; Nolan et al., 2008; Tascioglu et al., 2017).

The two models that are coupled in this study are both taken from previous works. The social ABM has been built on the above concepts and calibrated through thousands of training iterations (Harati et al., 2021a). The spatial ecological model has been developed, calibrated and tested with observed data (Harati et al., 2020). We refer the readers to these two papers for a detailed description of the models.

Complexities arise when the model's governing agent uses the '*responsible user*' label to encourage the user agents to engage in a costly action. At the beginning of the simulations, the label has not been introduced in the society of user agents and it is therefore not deemed valuable. Later on, as the label becomes more visible in society, its value increases in the calculations of the user agents. Meanwhile, an ecological disturbance causes damage to the forest resource. The model sheds light on the complexities of the above said setting. Specifically, the model helps answer these questions about the possibility of success

for the governing agent: Can the governing agent gain the cooperation of the user agents? Can they effectively control the disturbance? Can they save the resource?





The governing agent's decisions are based on bounded rationality (Simon, 1990). The governing agent does not know how the group of user agents behaves, it does not know what their decision thresholds are, and its information about the ecological system comes with a delay. The governing agent is designed in such a way that it observes the outcome of its actions, and learns to update its decision policy according to its observation. User agents make rational choices (Scott, 2000) based on information that is available to them. They do not modify their decision rule. User agents compare the utility of a suggestion with a threshold that indicates their hesitation, and make their decisions accordingly.

The social ABM uses input from the spatial ecological model. This input is the simulation of changes in a landscape, which is produced and processed through a GIS approach. This GIS approach does not take a time-series of external inputs during the simulations. However, the spatial ecological model is calibrated before the start of simulations using GIS data, which is available at grid cell level.

### A.2.2 Individual decision-making

Subjects of decision making are the governing agent and user agents. The object of decision making of the governing agent is its binary signal, which is a variable that the governing agent communicates to user agents. The signal indicates whether the governing agent is requesting a costly action or no-cost action from the user agents. The object of decision making of each user agent is its decision, which is the user's response to the governing agent's signal request.

Since the governing agent's decisions are based on bounded rationality and it does not have perfect knowledge of the complex system that it deals with, it takes actions according to its available knowledge. Then based on the result of its action, the governing agent updates its decision policy using a RL algorithm. The governing agent's RL algorithm is a double-learning algorithm, which means it includes two arrays of scores of *(state,action)* pairs. These two arrays are updated iteratively in a convoluted manner, each based on the other.

The user agents' decisions are based on rational choice. User agents calculate the utility of cooperating with the governing agent, and compare it with an internal decision threshold. The utility that user agents calculate is a quantity between 0 and 1. If the calculated utility of a suggestion exceeds a user agent's decision threshold then the user agent accepts that suggestion. In calculation of the utility of the *'responsible user'* label, user agents take into account the *uniqueness* that they will have with the label, and the *visibility* of being associated with responsibility. They assess *uniqueness* based on the last known proportion of user agents who cooperated with the governing agent in a costly action. They assess *visibility* based on the total number of times the label has been presented in their society since the first time step of the simulation. User agents calculate *uniqueness* and *visibility* based on the registrar's *nLast* and *nSum*.

User agents adapt to changes in their social and ecological environment. Social changes influence each user agent's perceived value of being recognized as a '*responsible user*', and ecological changes influence the size of the area where the management action is prescribed, hence influencing the cost of action required to receive the '*responsible user*' label. These variables do not change the decision rule of the user agent. The simulations shed light on the emergence of a norm of environmentally responsible behavior among user agents. On the other hand, the spread of infestations in the forest is a spatial process, which





influences the governing agent's perceived state of forest health, and subsequently, cost of management action in each user zone.

All agents in the model use memory in their decisions. User agents refer to the registrar's memory. The governing agent, in addition to the memory of the registrar, uses its own built-in memory. The governing agent's RL algorithm applies a future discounting rate in the calculation of the present value of future consequences of its decisions.

The model includes some elements of uncertainty. The decision thresholds of user agents are taken from a normal distribution. The decision policy of the governing agent is defined stochastically. That is, for each *(state,action)* pair, the policy includes a number, which is used as a threshold for comparison against a random number. The decision is made according to that comparison.

### A.2.3 Learning

Learning is the basis of the governing agent's RL algorithm. The RL algorithm keeps track of its states, actions, and rewards. The algorithm uses a policy to decide an action in each state. Then, based on the subsequent reward, the RL algorithm updates its policy. Through iterations, the governing agent's RL algorithm learns to adjust its policy in order to maximize its rewards. The model does not include collective learning.

### A.2.4 Individual sensing

In this model, individuals are the agents in the social ABM. The model includes endogenous and exogenous variables. As for endogenous variables, user agents sense the governing agent's signal. User agents and the governing agent sense the total number of '*responsible user*' labels as well as the last known number of labels given in a time step when signal was 1. These variables are accessible to agents through the registrar. These endogenous variables are sensed without error. As for exogenous variables, the governing agent senses the changes that happen in the ecosystem. In our conceptual model, these changes are

simulated by a spatial ecological model that is coupled to the social ABM. Therefore, this information is exogenous to the ABM. The sensing of environmental change is erroneous because in the definition of the model, environmental changes are not visible when they occur. The time lag between occurrence and visibility of the changes causes errors in the governing agent's sensing, thus adding to the complexity of the SES model. The governing agent and user agents sense the variables stored in the registrar, which is an auxiliary agent created for better understanding the model. The registrar, in turn, senses the

governing agent's signal and each user agent's decision. These variables are sensed without error. The governing agent senses the spatial environment at global and local scales, when it calculates the overall state of health of the forest and the cost of management action in each user agent zone, respectively.

Within the social ABM, when the governing agent, users agents, or the registrar require information from another agent, they call that other agent. The agents are equipped with functions that send the requested information. Agents do not have direct

access to variables of other agents. In the link between the social ABM and the spatial ecological model, each model is designed to perform some calculations, then wait for the other model to send the required information. This information is transferred



by copying a file into the recipient model's inbox directory. The model does not assume any costs associated with cognition or for gathering information.

### A.2.5 Individual prediction

The governing agent's RL algorithm uses the data gained through experience in order to assess the values of its possible actions in the next step. The user agents consider data of the last known states in their calculations. The governing agent uses a temporal difference RL algorithm known as Double Expected SARSA (Sutton and Barto, 2018). The user agents assess the future value of obtaining the '*responsible user*' label with the assumption that the agents who previously chose a costly action in return for a label, will do so again. The predictions of the agents may be erroneous. User agents have limited ability to predict future

changes in their society. Likewise, the governing agent's social prediction capability is limited. In addition, the governing agent's external input, which comes from the ecological spatial model, is designed to come with a delay.

### A.2.6 Interaction

The model includes direct and indirect interactions among agents. Direct interactions include the communication of governing agent's signal and action cost calculations, as well as user agents' decisions. Indirect interactions occur due to user agents'

desire to be better recognized than their peers, as well as through the market where all user agents sell their harvest. The governing agent's decisions and calculations depend on the history of responses from the user agents as well as the state of the ecological system. User agents' decisions depend on action costs, which are calculated through a spatial analysis. Interactions within the social ABM are communicated via the registrar. Interactions between the social ABM and the ecological spatial model are performed via file transfers, in which messages are copied into the recipient's inbox directory. The model does not

include a coordination network.

### A.2.7 Collectives

There are no collectives in this model.

### A.2.8 Heterogeneity

User agents are heterogeneous in their decision thresholds, as well as their allocated forest zones. User agents and the governing

agent are different in their decision making. The object of decision of the governing agent is the signal it sends to the user agents, and the objects of decisions of user agents are their responses to the governing agent. The governing agent uses a RL algorithm in its decision, whereas user agents compare the utility of a suggestion with a threshold.



### A.2.9 Stochasticity

The decision thresholds of the users are drawn from a normal distribution. The decision policy of the governing agent is
stochastic.

### A.2.10 Observation

In each time step, the governing agent's signal, the proportion of user agents who cooperated with the governing agent, and
remaining proportions of infested, non-infested, and harvested forest land are collected for analysis. In addition, for testing
and verification of the model, all communications between the social ABM and the ecological spatial model are saved. Among
the user agents, cooperation with the governing agent despite its cost is a behavior that emerges through simulations. In
addition, saving forest areas from infestations is an emergent effect in the simulations.

### A.3 Details

### A.3.1 Implementation details

The social ABM was developed in Java, using features of REPAST (North et al., 2013). The spatial ecological model was
developed in R (R Core Team, 2019). Please see the 'Data and code availability' section for links to model code and results.

### A.3.2 Initialization

The social ABM consists of a governing agent, nine user agents, and a registrar agent. The governing agent's policy is defined
by the results of a previous study (Harati et al., 2021a), wherein the governing agent's RL algorithm was trained through
interaction with the same number of user agents. There is no history of decisions of user agents, therefore the last known
number of user agents cooperating with the governing agent is zero. In the previous study that defined the social ABM (Harati
et al., 2021a), three sets of simulations were run with mean user agent decision thresholds of 0.3 (low), 0.5 (medium) and 0.7
(high). In the present study, assuming that the governing agent does not have any information about the decision thresholds of
user agents, we initialized the governing agent with the policy obtained from training with medium level user agent decision
thresholds. The said training was the output of the previous study (Harati et al., 2021a). In the spatial ecological model,
locations of insects at the start of simulations are extracted through GIS analysis of infestation data (BC Ministry of Forests,
2015).

There are some differences between various runs of the same series of simulations. The decision thresholds of user agents in
the social ABM are drawn from a truncated normal distribution with pre-set mean and standard deviation. In new runs, new
thresholds are drawn from the same normal distribution. Therefore, user agents change in new runs. For the governing agent,
within the same set of simulations, the decision policy is updated based on rewards earned in the previous episode of runs.





### A.3.3 Input data

In each time step, the social ABM uses input from the ecological spatial model. This input is based on simulations of spread of infestations in the forest. As the simulated infestations spread further in the forest, the data transmitted to the social ABM changes over time.

### A.3.4 Submodels

The social ABM includes a RL algorithm for the governing agent and a simple threshold decision-making algorithm for user agents. These are explained in detail in a previous work (Harati et al., 2021a). The ecological spatial model is based on a logistic regression algorithm, which is explained in detail in another previous work (Harati et al., 2020). The social and ecological models are coupled through a mechanism that we call flip-flop. The social model requires inputs about the state of

the forest and newly spread infestations, which is calculated in the ecological model. Conversely, the ecological model requires inputs on actions of user agents, which change land cover. In the flip-flop mechanism, each of the models runs its algorithm up to the moment it requires input from another model. Then it enters a loop in which it waits and observes an inbox directory that is allocated to that model in the computer's hard disk. In the meantime, the other model continues its calculations and eventually produces an output message file and sends it to the above-mentioned inbox directory. As soon as the message file

is copied into the inbox directory, the first model notices the change in the contents of its inbox, exits the waiting loop, reads the file and resumes computing. In this way, models take alternative turns of running and pausing, hence the name 'flip-flop'. This strategy has enabled us to facilitate the exchange of information between two different algorithms (i.e. the social and the ecological models) that have been implemented in two different computer languages (Java and R, respectively).

The management action that we consider in this study is cutting cells in a neighborhood of newly observed infestations. The

size of this neighborhood is a parameter that needs to be defined. Based on insight obtained about spatial spread of MPB infestations in a previous study (Harati et al., 2021b), in the present study we used Moore neighborhoods of size 4 to simulate the above-said management action. Considering that the cell-size in the model is 400 meters, the said neighborhood will be a square with side length of 3.6 kilometers. The rationale for this hypothetical neighborhood is that newly infested cells are not immediately detected. By the time infested cells change color and become observable, the infestation spreads further in the

area.

The subject of calibration of the social ABM is the decision policy of its RL algorithm. This policy was learned previously (Harati et al., 2021a) through 50 sets of 4000 training episodes each in a configuration with medium-level decision thresholds for user agents. Each set of 4000 episodes resulted in one (1) learned policy, thus, there were a total of fifty (50) learned policies. The mean of those 50 policies was used as the starting policy in the simulations of the present study. The spatial

ecological model was calibrated using observed infestation data of years 2002-2004 for BC. Details of the model and its calibration are described in the corresponding previous work (Harati et al., 2020).





As both the social ABM and the spatial ecological model are taken from previous works, we only made modifications to code and parameters for the coupling of the two models and the runs of this study. The social model's parameters include the number of user agents, mean and standard deviation of decision thresholds of user agents, future discounting rate, number of time steps in one episode, and number of episodes. Each episode of simulations was run with a new set of user agents. In addition, in the present study we added a new parameter for the number of preparation steps before the social model is coupled with the spatial ecological model. The parameters of the coupling of the two models are the business-as-usual harvest ratio, which is the ratio of the study area that the user agents would harvest regardless of disturbance management, and the size of the neighbourhood of newly visible infestations, in which the management action of cutting cells is defined. Table A.2 shows the model parameters and their values. Note that in this table all values are dimensionless except for the management action neighborhood size, which is in grid cells.

**Table A.2: Model parameters.**

| Parameter | Value(s) |
|---|---|
| Number of user agents | 9 |
| Mean decision threshold of user agents | 0.7 |
| Standard deviation of decision thresholds of user agents | 0.08 |
| Future discounting rate | 0.1 |
| Number of time steps in one episode | 10 |
| Number of preparation time steps | 0, 10, 20 |
| Number of episodes | 50 |
| Business-as-usual harvest ratio | 0.01 |
| Management action neighborhood | Moore, size 4 |

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
