# Peer review of "Learning from conceptual models – a study of emergence of cooperation towards resource protection in a social-ecological system"

_Geoscientific Model Development, 2024_

## Author Response (AR1)

**Reviewer1**

We are thankful to the respectable reviewer for these comments, which helped us improve the manuscript. The revised manuscript will be uploaded upon approval of the discussion by editors. Here are our responses to the reviewer's specific comments:

1. On modelling choices

- In this particular context, we did not find works in related literature that address the effect of recognition on emergence of environmentally responsible behavior in logging companies. Our work in this sense is a modest contribution to the field.
- The reviewer raised an important question on the justification for simple models in the study of complex systems. In the case of our social model, our aim was not to produce exact predictions, but rather to gain insight. For us, a simplified and abstract modelling approach was appropriate, as it allowed us to see how model configurations affect its output. This would have been very difficult if we had started our model with multiple degrees of freedom, because complex systems may reach the same state from various paths of change. With the abstract model, we now have an insight about the emergence of our intended behavior. Of course, in reality such emergence may be hindered or hastened by other factors that we have not considered. But still it in insightful to see how the system under study tends to evolve in absence of those other factors.

2. On clearer description of the social model

- As the reviewer correctly noted, our model's "governing agent" is the entity that is referred to as "principal" in "principal-agent problem" literature. We understand that a confusion arose because the term "agent" has different meanings in "agent-based modelling" on the one hand, and in "principal-agent problem" literature. We add a clarification on this matter in the revised manuscript.
- In the first draft of the manuscript we referred the readers to our previous work for details on the decision making of user agents. As per the reviewer's suggestion, we can add these details in the revised manuscript. As the reviewer correctly noted, these decisions are based on simple if-statements. User agents seek uniqueness *and* value in their actions. To that end, they assess scores for expected uniqueness and value, then they multiply those scores. Numerical multiplication here performs as the logical "and" operator. Assessment of uniqueness is based on the immediate past. For example, if no other user showed responsible behavior in the previous time step, a full score of uniqueness is assumed for the respective action in the present time step.
Assessment of value is based on the cumulative past. It represents the total number of times the "responsible user" label has been seen in the society. That is the number of times the label has been awarded to agents.
Each user agent has a numerical threshold that it compares with the product of uniqueness and value in each time step. The result of that comparison defines the user agent's decision to act. The said threshold varies from one user agent to another, but it is constant for each user agent throughout simulation.

- We suggested the possibility of adding an economic model as an avenue for future work. Presently our model does not have an economic component. We modify the wording of our text in the revised manuscript to avoid confusion.

3. On moving parts of text from the discussion section to the methods section

- We thank the reviewer for this comment. Our initial intention was to present background information for discussion, but we find the reviewer's suggestion improves the flow of the text better. We therefore revise the manuscript accordingly.

4. On what happens during preparation time in simulations

- During the preparation time, the social model runs alone, before starting the ecological model. This can represent awareness raising campaigns. During this time, the "responsible user" label is introduced to the society, and user agents get a chance to compete for recognition. Through this competition, the "responsible user" label becomes more visible, and therefore its value increases. After the preparation time, the social and ecological models are coupled together, and the social model keeps its memory of the value of the "responsible user" agent. That is why even in our "random" scenarios, where the governing agent does not learn, there is some action by user agents. This shows the importance of the desire for recognition, which creates a strong potential for emergence of environmentally responsible behavior. Such potential, of course, is not optimally used when the governing agent's decisions are random.

5. On repetitions in the results part

- We appreciate the reviewer's comment. In the results section we had the challenge of presenting multiple scenarios with similar configurations. In the end, we decided to keep the descriptions of scenarios and results long and complete, to avoid the risk of confusion and misinterpretation.

Technical corrections

- We thank the reviewer for this observation. Wrong figure numbers were mentioned in the text (5 and 6 instead of 6 and 7). Corrections are applied in the revised manuscript.

Reviewer 2

We are grateful to the respectable reviewer for these comments, which helped us better present our work. The revised manuscript will be uploaded upon approval of the discussion by editors. Here are our responses to the reviewer's specific comments:

1. On modelling governance agents

- The reviewer correctly noted the difference between real-world policy making and our model. Our model does not aim at producing exact replications or predictions of real world. Rather, the objective of our model is to provide insight about a complex systems problem. As the reviewer correctly mentioned, a constraint of real-world governance situations is lack of sufficient data for policy optimization. Repeated trial-and-error experiments are not possible in real-world situations. Exactly to address this constraint, our approach has been to develop a virtual laboratory where we can run experiments without risk of adverse effects. Such abstraction allowed us to overcome real-world constraints and produce datasets that could be analyzed for optimum policy selection. Then, the advantage of reinforcement learning was accessible, as it allows to associate observed rewards with sequences of past actions. In summary, our abstract reinforcement learning model allowed us to run experiments that were otherwise not possible in the real-world.

2. On presenting novelty

- We thank the reviewer for this substantial comment. Before discussing the reviewer's point, we add as a side note that this work and the previous publications describing our social and ecological models comprise the first author's doctoral research. Indeed, the studies of our previous publications were carried out exactly in order to make the present work possible.
- The reviewer correctly noted that the "responsible user" label is a key factor influencing simulation outcomes. Here we find it necessary to emphasize the distinction between our social model (previous publication) and the present social-ecological model. In our previous publication, which describes the social model, the decision parameters of the user agents were fixed. Specifically, each user agent's perception of the cost of requested actions was a fixed value throughout the simulation. In contrast, in the present work, user agents' perceptions of the cost of requested action depend on the volume of infestation in their allocated forest zones, which varies depending on not only ecosystem dynamics, but also depending on actions of users. As infested areas increase, creating buffer zones around them becomes more costly. Subsequently, user agents become less motivated to participate in the management action of creating buffer zones to stop spread of infestations. This is an added layer of complexity that distinguishes the present social-ecological simulations from the social simulations of our previous publication.
- As the reviewer mentioned, our previous publications addressed technical challenges regarding the social and ecological models of our study. The present study involved technical challenges too, such as the coupling of the models and the analysis and interpretation of results. Nevertheless, we would like to emphasize that the present study

additionally addressed the challenge of translating a complex situation into a problem definition, and subsequently developing an approach for that problem. This is especially important in the domain of decision support for sustainable development. Without a well-problem formulation and without an understandable approach, decision making in sustainable development will rely on individuals' subjective perceptions of subject matters, which are often complex and multifaceted. We demonstrated an exercise of formulation of the problem and an approach of dividing the problem into smaller parts, conquering each part through model development, and finally re-integrating the modelled parts. We hope that our work serves decision making by providing a more formal and less subjective ground for developing and discussing ideas.

Consider for example that the scope of decision making about our social-ecological system is to become wider by considering an additional aspect – market and economic complexity. Arguably, this would be a complex case where intuition does not provide clear insight on how the interaction of system components evolves. Respectively, it would be challenging to make decisions about intervention in such a complex system, primarily due to lack of insight. Our modelling approach simply allows to develop an independent market model of timber supply and demand, and subsequently couple that to the existing social-ecological model by adding an expected revenue term to the cost calculations of user agents.

3. On the methods section

- We thank the reviewer for this comment. We find the reviewer's comments improves the text, and we apply it in the revision of the manuscript.

4. On Figure 3

- We thank the reviewer for this comment. We use the reviewer's suggestions to improve our flowchart in the revision of the manuscript.

Technical corrections

- We thank the reviewer for this comment. Indeed, our effort to summarize the legend only eliminated one category in six. We find the reviewer's suggestion improves the figure, and we apply it in the revision of the manuscript.

---

## Author Response (AR2)

*We have applied the specified minor revision in the manuscript. Please find our responses in italics inline with your comments below. We have identified line numbers of revised parts of the manuscript in the "Author's track-changes file" in the system.*

**Public justification (visible to the public if the article is accepted and published):**

Thank you for your responses to the reviewers' comments. I have some responses of my own for which I am requesting minor revisions:

- Line 551: "lead" should be "led".

*+ We corrected this mistake. Please see line 436 of the revised manuscript.*

- Please mention in the manuscript that there is no economic component, which might be unintuitive to readers, as it was for Reviewer 1.

*+ We applied this comment in the revised manuscript. Please see lines 119-120.*

- Your response to Reviewer 2's first point is a good one. I think it's worth explicitly including in the Discussion (maybe at the end of Sect. 4.3) both their note that "policymakers do not have access to data-intensive training opportunities to optimize their strategies" as well as your response that that is why your model is useful—to inform policymakers.

*+ We added a new paragraph addressing this comment. Please see lines 438-446.*

- Figures 3 and 5 appear unchanged from the original manuscript. Did you forget to update them as you said in response to Reviewer 2?

*+ We inserted new versions of Figures 3 and 5 in the revised manuscript.*

*+ Also, as requested by the journal, we merged the two reference lists into one, at the end of the main text.*